# A Novel Changing Athlete Body Real-Time Visual Tracking Algorithm Based on Distractor-Aware SiamRPN and HOG-SVM

**Mingwei Sheng** [1,2], **Weizhe Wang** [1], **Hongde Qin** [1,*], **Lei Wan** [1], **Jun Li** [1] and **Weilin Wan** [2]

[1] Science and Technology on Underwater Vehicle Laboratory, Harbin Engineering University, Harbin 150001, China; smwsky@163.com (M.S.); wangweizhehit@163.com (W.W.); wanlei@hrbeu.edu.cn (L.W.); lijun_k@163.com (J.L.)

[2] Department of Computer Science, The University of Hong Kong, Hong Kong 999077, China; wanwl@hku.hk

[*] Correspondence: qinhongde@hrbeu.edu.cn

**Abstract:** Athlete detection in sports videos is a challenging task due to the dynamic and cluttered background. Distractor-aware SiamRPN (DaSiamRPN) has a simple network structure and can be utilized to perform long-term tracking of large data sets. However, similarly to the Siamese network, the tracking results heavily rely on the given position in the initial frame. Hence, there is a lack of solutions for some complex tracking scenarios, such as running and changing of bodies of athletes, especially in the stage from squatting to standing to running. The Haar feature-based cascade classifier is involved to catch the key frame, representing the video frame of the most dramatic changes of the athletes. DaSiamRPN is implemented as the tracking method. In each frame after the key frame, a detection window is given based on the bounding box generated by the DaSiamRPN tracker. In the new detection window, a fusion method (HOG-SVM) combining features of Histograms of Oriented Gradients (HOG) and a linear Support-Vector Machine (SVM) is proposed for detecting the athlete, and the tracking results are updated in real-time by fusing the tracking results of DaSiamRPN and HOG-SVM. Our proposed method has reached a stable and accurate tracking effect in testing on men's 100 m video sequences and has realized real-time operation.

**Keywords:** visual tracking; real-time; Siamese tracker; HOG-SVM

## 1. Introduction

Nowadays, it is possible to obtain various types of videos through various channels, and automatic video analysis is becoming a quite urgent task. Since it is time consuming and labor intensive to analyze sports video manually, automatic sports video analysis has received much attention, which involves disciplines such as image processing knowledge, pattern recognition, and artificial intelligence. Therefore, application of automated video analysis has been widely used in motion analysis, video surveillance, and athlete tracking [1,2].

People keep healthy and strong through sports such as basketball and football. Running is the basis of most sports, and has been an important part of competitive sports in the Olympic Games. Over the past couple of decades, the technical level of running training has developed slowly. Subjective and experience-based teaching methods have been used. Coaches detect the technical movements of athletes with their naked eyes and experiences. After long-term practice and research, sports experts believe that the introduction of digital video technology in sports training is conducive to significantly improving training efficiency. Therefore, it is necessary to conduct automatic analysis of sports videos, such as detecting and tracking the motion parameters of sports videos to analyze the behavioral characteristics. Equipping athletes with various sensors is a traditional detection and tracking method

for quickly and accurately obtaining useful information [3]. However, it brings additional burdens to the athletes and seriously affects their competitive level.

Video-based detection and tracking methods for athletes, as non-contact methods, have been widely studied and applied in fields such as basketball [4–6], football [7–9], and volleyball [10]. Athlete detection and tracking is the basic element for understanding the competition. In general, there are some difficulties in automatic athlete detection, such as the squatting start in a running race that affects subsequent prediction, as well as athletes with similar appearances, complex interactions and large-area occlusion, various outdoor environments, changing backgrounds, unpredictable numbers of athletes, unpredictable movement, sudden movement of the camera, zooming, and calibration. The detection and tracking of athletes also face great challenges due to the low texture area, broadcast video editing, noise, insufficient pixel resolution (especially when playing at small distances), clutter, and motion blur that result in inaccurate results.

Inspired by the above studies, this paper implements the Haar feature-based cascade classifier for detecting the key frame representing the dramatic changes of athletes, and improving tracking accuracy by combining Distractor-aware SiamRPN (DaSiamRPN) and Histograms of Oriented Gradients with a linear Support-Vector Machine (HOG-SVM) to realize real-time tracking. The main contributions of our work are as follows.

1. The dramatic body changes of athletes during a running race have a great influence on athlete tracking. DaSiamRPN has achieved a leading position in real-time evaluation, but it heavily relies on the given position in the initial frame. In order to reduce the influence of body changes on tracking results, DaSiamRPN is involved for generating a based bounding box. Therefore, a combined tracking method of DaSiamRPN and HOG-SVM based on key frame is proposed in order to detect and track the athletes during the race.

2. A cascade classifier based on the Haar feature is implemented for catching the convert key frame. During the running race, the height and width of the bounding boxes of athletes identified by cascade classifier are calculated to determine the key frame, which represents the frame in which the athlete's body changes the most dramatically. This method not only helps to reduce the tracking error by DaSiamRPN, but also helps switch to a new tracking mode.

3. We propose a novel tracking method utilizing the DaSiamRPN and HOG-SVM athlete tracking results at the same time. Before the application of HOG-SVM, we trained thousands of positive and negative samples of athletes. The proposed algorithm can further enhance the adaptability of dramatic changes in athletes' bodies. Experiments on actual race sequences indicate that our proposed algorithm can realize accurate tracking of athletes in real-time.

The rest of this paper is organized as follows: Section 2 briefly reviews previous research works and related technologies. Section 3 details the Haar feature-based cascade classifier, DaSiamRPN-based HOG-SVM Tracker, and the fusion tracker of DaSiamRPN and HOG-SVM. Section 4 evaluates the proposed tracker's performance and compares the proposed algorithm with other trackers; then, the experimental results are discussed. Section 5 presents the conclusion and future prospects.

## 2. Related Work

Two key points of the research method are the Distractor-aware SiamRPN and HOG. Video target tracking has become a hot topic, and several deep-learning-based trackers have been applied in multiple fields. However, target changes caused by deformation, occlusion, and motion are still major issues that cannot be ignored [11,12]. It is a common problem for most autonomous driving and video surveillance systems [13,14] that the tracker lacks real-time performance [15,16].

In recent years, one of the important research directions in the tracking field is tracking algorithms based on correlation filtering, such as Kernelized Correlation Filters (KCF) [17], (Scale Adaptive with Multiple Features tracker) SAMF [18] and MUlti-Store Tracker (MUSTer) [19]. These algorithms have better performance in tracking accuracy and speed. The feature expression of the target is an important factor affecting target tracking performance. In practical applications, it often faces

complex environments. Traditional artificial feature detection is insensitive to the shape change of the target. Deep networks have remarkable performance in feature expression. Luca Bertinetto proposed a fully convolutional twin network based on similarity learning, SiamFC [20] (fully convolutional Siamese network) and a tracking-based correlation filter network, CFNet [21] (correlation filter based tracking), which introduces a correlation filter onto shallow features. The twin instance tracking network, SINT [22] (Siamese instance search for tracking), combines optical flow information on the basis of the twin network to achieve better performance. Although the above tracking algorithms realize significant improvements in accuracy, when the target is deformed or occluded, it is still prone to drift, and thus results in a decrease in algorithm accuracy. Danelljan et al. [23,24] proposed a method of generating models based on the deep features of correlation filters, which increases the diversity of training samples, optimizes the objective function, and improves accuracy. However, it is complex and has a low tracking rate. Liu et al. [25] proposed a novel template-matching tracking algorithm. The algorithm obtains the most accurate results from previous tracking results by using k-nearest neighbors, but it simply uses a simple machine learning algorithm to classify samples, resulting in poor performance and low accuracy. However, SiamFC simply pays attention to the color similarity of the target, so only shallower features are obtained. The size of the marker box does not change; for example, when the target in the video moves from far to near, the marker box will not grow as the target gets larger. In the presence of interference from other objects, it is easy to detect the wrong objects because of the single datum for feature extraction.

SiamRPN is a combination of Siamese Network and Region Proposal Network (RPN), which proposes an end-to-end offline training method and regards the tracking process as one-shot detection. The implementation of SiamRPN on relevant feature maps has achieved a leading position in real-time evaluation, which proves its advantages in accuracy and efficiency [26]. Specifically, SiamRPN consists of a conjoined sub-network for feature extraction and a regional suggestion sub-network including classification and regression branches. The previous part of SiamRPN is the same as SiamFC: It first extracts high-level features through a full convolutional network. The difference is that SiamFC directly utilizes the output for correlation filtering, while SiamRPN accesses RPN. As a SiamRPN-based algorithm, DaSiamRPN mainly aims at training data, enriches the number and type of samples, makes the tracker more robust by utilizing better training methods, and proposes local-to-global ideas for long-term tracking problems.

Histogram of Oriented Gradient (HOG) constructs features by calculating and counting the gradient-direction histograms of local image area, and is usually utilized for object detection in computer vision and image processing. Compared with other characterization methods, HOG has many advantages, and it is particularly suitable for human detection. Firstly, HOG operates on the local grid cells of the image, so it can keep invariance to deformation, including the geometric and optical deformations in a larger spatial domain. Secondly, fine directional sampling and strong local optical normalization are allowed to have some fine limb movements under the conditions of coarse spatial sampling. These fine movements can be ignored without affecting the detection effect. Furthermore, the HOG combined with an SVM classifier (HOG-SVM) has been widely implemented in the image recognition field, especially in pedestrian detection [27].

Su et al. [28] presented an SVM-based tracking method applying the loop structure of samples to process the first and second factors through the multi-core learning mechanism. Specifically, an SVM classification model for visual tracking was developed. This model combines two types of matrix loop kernels, and makes full use of the complementary features of color and HOG features to learn robust target representation. Pang et al. [29] proposed a unified network (called JCS-Net) for small-scale pedestrian detection based on HOG + LUV and JCS-Net, which constructs a multi-layer channel feature (MCF) to train detectors. Qu et al. [30] proposed a machine learning algorithm for texture information extraction from through-focus scanning optical microscopy (TSOM) images. Compared with the results of the library matching method, the machine learning method has a much higher measurement accuracy. However, HOG also has some disadvantages. Its descriptor generation process

is lengthy, resulting in problems such as its slower speed, inability to handle occlusion, sensitivity to noise, and proneness to error tracking.

Therefore, in order to avoid the problems caused by athletes' follow-up squat-starting methods, such as sudden changes in follow-up calibration frames, tracking failure, and over-dependence of SiamRPN on the initial frame during the starting process, this paper proposes the use of the DaSiamRPN and HOG methods for tracking. The accuracy and real-time advantage of the upper bodies of athletes are matched with the trained model of HOG-SVM to detect the enlarged frame.

## 3. Proposed Tracking Based on Distractor-Aware SiamRPN and HOG-SVM

### 3.1. Proposed Tracking Algorithm Structure

In this section, a novel detection and tracking framework combining distractor-aware SiamRPN and HOG-SVM is proposed, which can track the dramatic changes of athletes' bodies in running. To reduce the tracking failure caused by body changes, the key frame is defined as the important stage. The Haar feature-based cascade classifier is involved to obtain the bounding box of each athlete, and the height/width ratio is taken as the representation of the dramatic change stage, from squatting to standing to running. The proposed tracking algorithm structure consists of two stages. In the first stage, DaSiamRPN is implemented to track athletes, and the bounding box is viewed as the tracking results until the key frame is found. Then, a DaSiamRPN-based HOG-SVM tracker is proposed to detect the athlete. The results of DaSiamRPN and HOG-SVM tracking will be combined, and the fusion bounding box will be taken as the tracking result. The overall architecture of the proposed algorithm is detailed in Figure 1.

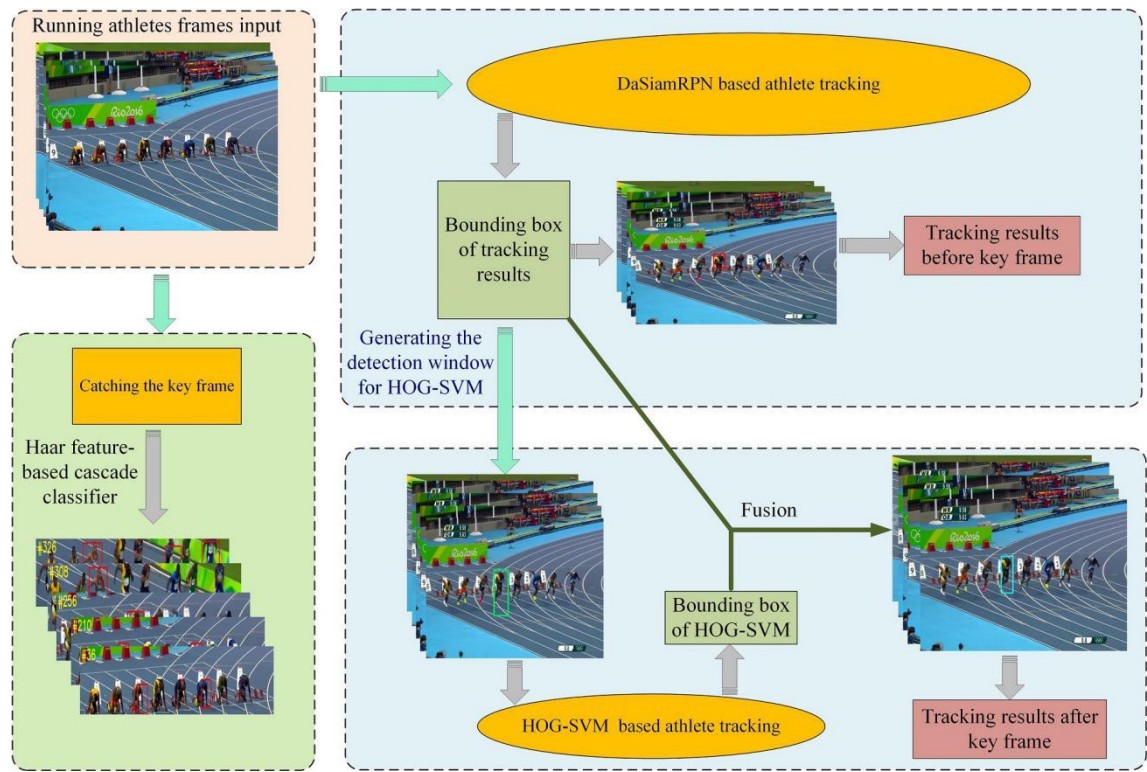

**Figure 1.** The architecture of proposed detection pipeline tracking combining the distractor-aware SiamRPN (DaSiamRPN) with Histograms of Oriented Gradients with a linear Support-Vector Machine (HOG-SVM).

### 3.2. Haar Feature-Based Cascade Classifier for Catching the Convert Key Frame

To detect the changes of athletes in running, a Haar feature-based cascade classifier is utilized. Each Haar-like feature is calculated by subtracting the sum of pixels under the white rectangle from the sum of pixels under black rectangle, which results in a single value [31]. Haar-like features are represented by the rectangle features for fast human body detection, as shown in Figure 2. To detect the athlete through the Haar-like features, the image is scanned from its top left to bottom right corner for several times.

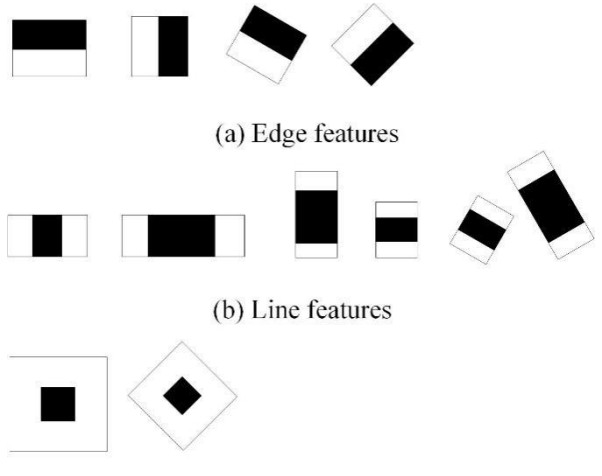

**Figure 2.** Examples of Haar-like features.

The integral image refers to the pixel values of the input image, which is used for fast feature detection. It is mainly utilized to increase the operation speed of the box filter. The athlete detection can be implemented by cascade-adopting Haar-like features. In a cascade, an image is assumed to include athletes if it passes all stages, or to have no identified athlete if it fails in any stage. The Open Source Computer Vision Library (OpenCV) [32] provides a training method or pretrained models, and the necessary haarcascade_fullbody.xml file is loaded by adopting the cv::CascadeClassifier::load method. Therefore, we directly used the cv::CascadeClassifier::detectMultiScale method to detect athletes in sequence frames, which returns bounding boxes for identified athletes.

However, to track the dramatic changes of athletes in running, we used a bi-axial scale change in two-dimensional space with the maximum value of change as the key frame. As the detection results of the Haar feature-based cascade classifier, each athlete detected is given a bounding box, and the aspect ratio of the bounding box in the same frame is calculated as

$$R_i^t = \frac{h_i^t}{w_i^t} \tag{1}$$

where $i$ represents the serial number of the identified athlete in the same frame, and $t$ represents the serial number of each frame. $h_i^t$ and $w_i^t$ represent the height and width of the bounding box, respectively.

To eliminate accidental errors, the average ratio $B^t$ is formulated as:

$$B^t = \frac{\sum\limits_{i=1}^{n} R_i^t}{n} \tag{2}$$

where $n$ is the digital frame that includes the athletes identified by the Haar feature-based cascade classifier.

As shown in Figure 3, there are different postures and aspect ratios $R_i^t$ in different frames. Finally, we define the key frame by considering the number of consecutive frames, namely, the identified average ratio $B^t$ surpasses the threshold $\beta$ for $k$ times. The conversion key frame represents the stage from squatting to standing. In consecutive frames after the key frame is caught, we use the bounding box generated by DaSiamRPN as a benchmark to enlarge it as a detection window for HOG-SVM tracking.

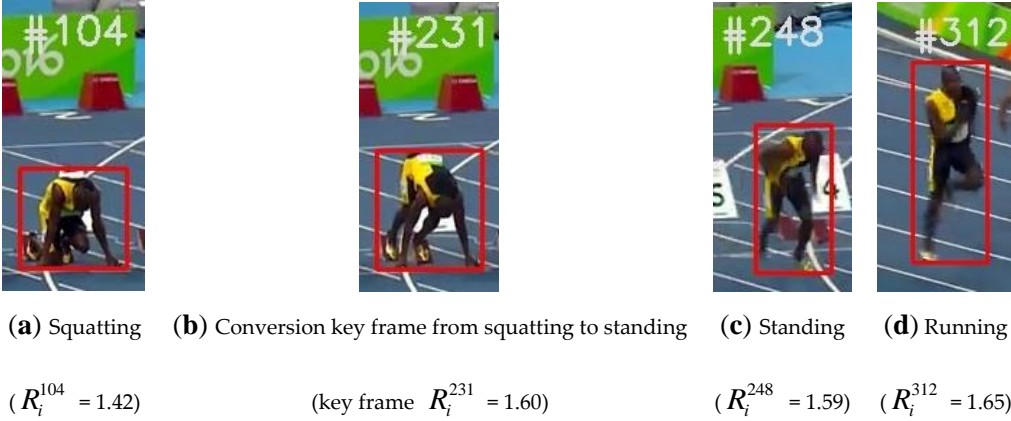

(**a**) Squatting　　(**b**) Conversion key frame from squatting to standing　　(**c**) Standing　　(**d**) Running

( $R_i^{104}$ = 1.42)　　　　　　　　(key frame $R_i^{231}$ = 1.60)　　　　　　( $R_i^{248}$ = 1.59)　　( $R_i^{312}$ = 1.65)

**Figure 3.** Postures during the race and the corresponding aspect ratios.

### 3.3. DaSiamRPN-Based HOG-SVM Tracker

We devised an HOG-SVM tracking method based on the DaSiamRPN tracking results. HOG-SVM tracking refers to a method combining HOG features and the linear SVM for tracking athletes. As shown in Figure 4, the height and width of the DaSiamRPN tracking results in a bounding box $B_o^t$ are assumed to be $h_d$ and $w_d$, respectively. Since the initial given bounding box of DaSiamRPN is the upper body of an athlete changing from squatting to running, DaSiamRPN only tracks the upper body in consecutive frames. To track the whole body of the athlete, we enlarge the bounding box $B_o^t$ with a scale parameter ($\alpha_d$, $\alpha_h$) and keep the upper left of the bounding box unchanged. Then, we use a new bounding box $B_o^t{'}$ as a detection window for body tracking. The HOG features present the rough shape of the athlete by generating the histogram of the luminance-gradient vectors in the detection window $B_o^t{'}$. A linear Support-Vector Machine (SVM) is used as the learning model and a discriminant function is used for pattern recognition.

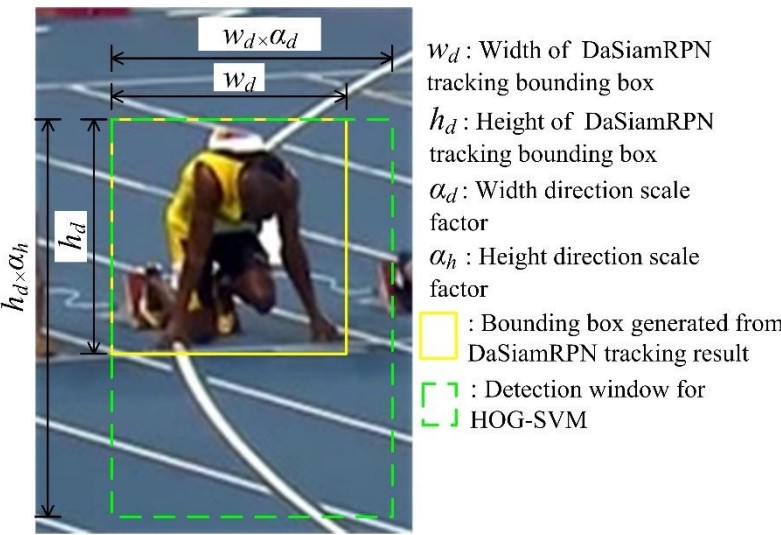

$w_d$ : Width of DaSiamRPN tracking bounding box

$h_d$ : Height of DaSiamRPN tracking bounding box

$\alpha_d$ : Width direction scale factor

$\alpha_h$ : Height direction scale factor

☐ : Bounding box generated from DaSiamRPN tracking result

⌐⌐ : Detection window for HOG-SVM

**Figure 4.** New detection window based on the DaSiamRPN tracker.

Figure 5 illustrates the whole image, detection window, block, cell, and image pixel. The whole image is captured as a frame from the sport video. The linear SVM can learn the parameters of the discriminant function; for instance, several images including the athlete Bolt were trained before the detection. The training results are saved as a myHogDector.bin file, and this file can be loaded by adopting the cv::hog::load method. Therefore, the cv::hog::detectMultiScale method is implemented to detect the athlete Bolt in the detection window and to generate a bounding box.

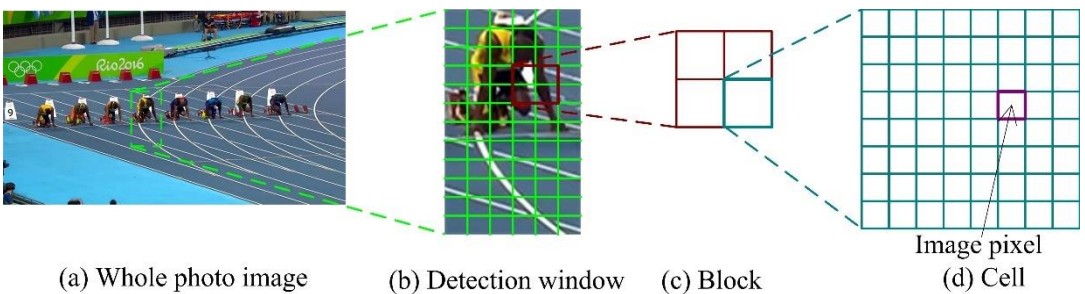

(a) Whole photo image　　　　(b) Detection window　　(c) Block　　　　(d) Cell

**Figure 5.** Whole image, detection window, block, cell, and image pixel.

### 3.4. Proposed Tracking Algorithm Based on the Combination of DaSiamRPN and HOG-SVM

The proposed tracker can be divided into two stages: In the first stage, the tracking result bounding box $B_o^t$ of DaSiamRPN is taken as the final tracking result bounding box $B_f^t$ method before the catching of the key frame. After the key frame is caught, the combined tracking algorithm is adopted. The tracking results of DaSiamRPN and HOG-SVM are fused in the combined tracking algorithm. The whole procedure of this method is given in Algorithm 1. The combined tracking bounding box is shown in Figure 6. As shown in Figure 6, the *o-xy* coordinate frame (Global coordinate frame) is fixed to the whole image.

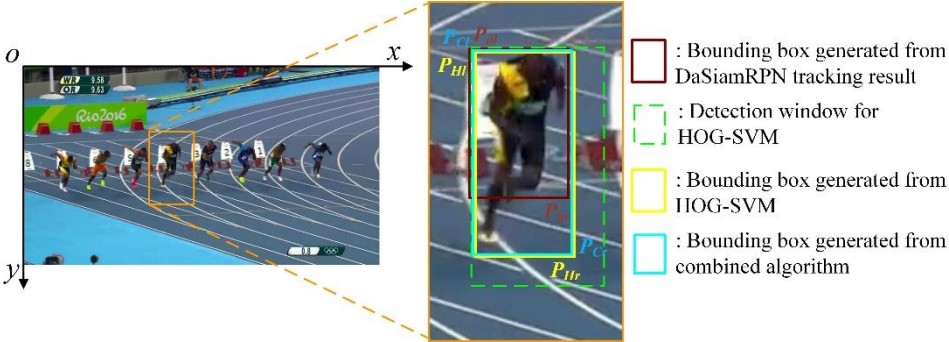

**Figure 6.** Combined tracking bounding box fusing the results of DaSiamRPN and HOG-SVM.

Therefore, the final tracking result $B_f^t$ can be generated by utilizing the combination of the results of DaSiamRPN and HOG-SVM. In the bounding box $B_f^t$, the top left corner point $P_{Cl}^t(x_l^C, y_l^C)$ and bottom right corner point $P_{Cr}^t(x_r^C, y_r^C)$ need to meet the following conditions:

$$\begin{cases} x_l^C = x_l^D \\ y_l^C = y_l^D \end{cases} \tag{3}$$

$$x_r^C = \begin{cases} x_r^D & if \quad x_r^D \geq x_r^H \\ x_r^H & if \quad x_r^D < x_r^H \end{cases} \tag{4}$$

$$y_r^C = \begin{cases} y_r^D & if \quad y_r^D \geq y_r^H \\ y_r^H & if \quad y_r^D < y_r^H \end{cases} \tag{5}$$

where $P_{Dl}^t(x_l^D, y_l^D)$ and $P_{Dr}^t(x_r^D, y_r^D)$ are the top left corner point and bottom right corner point of the DaSiamRPN tracking bounding box $B_o^t$, respectively. $P_{Hl}^t(x_l^H, y_l^H)$ and $P_{Hr}^t(x_r^H, y_r^H)$ are the top left corner point and bottom right corner point of the HOG-SVM tracking bounding box $B_{h'}^t$ respectively.

---

**Algorithm 1.** Athlete tracking with body dramatic change.

---

**Input:** $\mathbb{N} = \{N_1, N_2, N_3, \ldots, N_n\}$, video sequence frames
**Output:** $\mathbb{R} = \{B_o^t\}$, final tracking result sequence to frame $t$

1:    Initialization: $t = 0$, key_frame_state = 0, input an initial bounding box of the athlete
2:    **for** $t = 0$ to $n$ **do**
3:        Track the athlete in sequence $N_t$ using DaSiamRPN
4:        Predict a tracking result with a bounding box $B_o^t$
5:        $B_o^t$(top left corner point $P_{Dl}^t(x_l^D, y_l^D)$, bottom right corner point $P_{Dr}^t(x_r^D, y_r^D)$)
6:        Detect athletes in sequence $N_t$ utilizing the Haar feature-based cascade classifier
7:        Compute $R_i^t = h_i^t / w_i^t$ and $B^t$ as Equation (1), (2)
8:        **if** continuous $k$ frames with $B^t > \beta$ **then**        //means the key frame is found
9:            **return** key_frame_state = 1     //change to another tracking mode
10:       **else**
11:           **return** key_frame_state = 0     //no key frame is found
12:       **end if**
13:       **if** key_frame_state = 1 **then**       //the key frame is found
14:           Generate a detection window $B_o^t{}' = B_o^t *$ scale parameter
15:           $B_h^t = \text{HOG-SVM}(B_o^t{}')$
16:           $B_h^t$(top left corner point $P_{Hl}^t(x_l^H, y_l^H)$, bottom right corner point $P_{Hr}^t(x_r^H, y_r^H)$)
17:           Fusion of $B_o^t$ and $B_h^t$ as Equation (3), (4), (5), $x_l^C = x_l^D$, $y_l^C = y_l^D$
18:           $(x_r^C, y_r^C) = (\max(x_r^D, x_r^H), \max(y_r^D, y_r^H))$
19:           **return** the final result bounding box $B_f^t$
20:       **else**
21:           The bounding box $B_o^t$ is utilized as the final tracking result
22:           **return** the final result bounding box $B_f^t$
23:       **end if**
24:    **end for**     //reach the end of the video sequence

---

## 4. Experiments and Analysis

### 4.1. Experimental Setup

The testing dataset used included the videos of the men's 100 m race at Rio 2016 (Video Sequence I, including 510 frames) and London 2012 (Video Sequence II, including 380 frames) Olympic Games. We transferred the videos into 890 frames and implemented them with the help of an Intel(R) Core (TM) i7-9750H CPU, 2.60 GHz with 16 GB of memory and an NVIDIA GTX 1660 Ti GPU with 6 GB of memory. The driver version of the graphics card was NVIDIA UNIX x86_64 Kernel Module 384.81. The code was compiled by Python 3.6, and all code was performed on the operating system. The construction and testing of the network were implemented by Python 3.6 + keras 2.1.5 + tensorflow 1.9.0.

### 4.2. Search Results of the Key Frame

Figure 7 shows the person-tracking results on Video Sequence I using the algorithm of Section 3. As shown in Figure 7, although the OpenCV detection method cannot identify all of the athletes in all frames and the bounding box cannot cover the whole athlete's profile, the aspect ratio is approximate to the real. Therefore, a height/width scatter figure can be drawn, as shown in Figure 8. As depicted in Section 3.2, $\beta$ was set to 1.56 and $k$ was set to 8. Figure 8 shows that the frames surrounded by red dotted lines are the positions of the key frames of Video Sequence I.

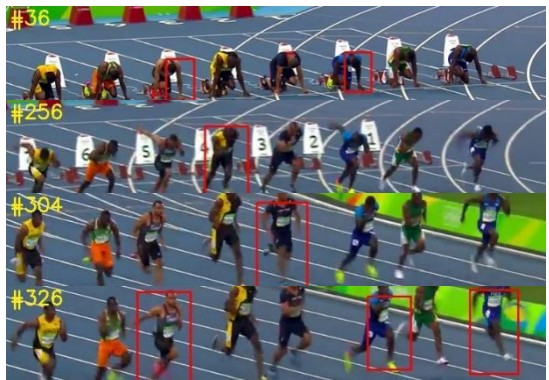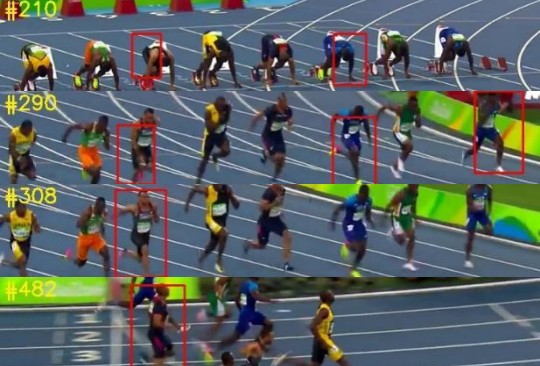

**Figure 7.** Athletes detection results of Video Sequence I.

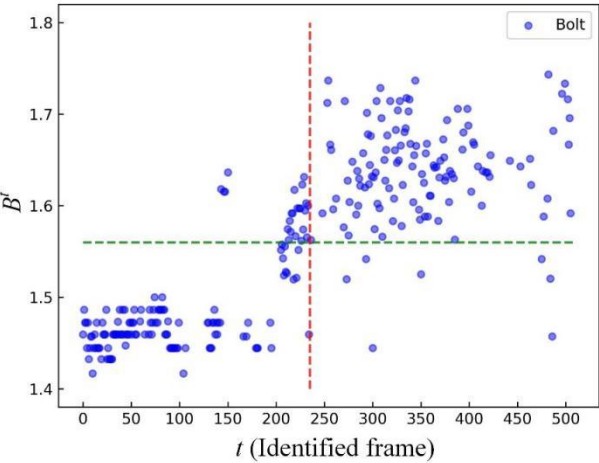

**Figure 8.** Scatter diagram of the average ratio $B^t$ of the bounding box height and width for each identified frame of Video Sequence I.

*4.3. Performance Evaluation*

We discuss the comparison with state-of-the-art tracking methods, such as the framework of DaSiamRPN, the combination of DaSiamRPN and HOG-SVM algorithm without considering the key frame, and the HOG-SVM algorithm on the Video Sequences.

The evaluation in this paper is based on two indicators: Success plots and precision plots. Figure 9 shows the overlapping precision scatter distribution diagram of the proposed algorithm compared with three other real-time trackers. As shown in Figures 9 and 10, the points with overlapping precision of less than the threshold of 0.5 are painted in red, and the other points are painted in green. Figures 9 and 10 demonstrate the visualizations of the final tracking results. As shown in Figures 9 and 10, the proposed algorithm obtaining the most points in green represents that the proposed tracker can achieve a stable tracking performance.

We also conducted ablation analysis to prove the effectiveness of the proposed algorithm. The average overlap precision (OP) is defined as the percentage of frames in the video where the intersection exceeds the threshold of 0.5. The area under the curve (AUC) is calculated from the success plots, where the average OP of all frames is plotted within the threshold [0, 1]. The precision plots and success plots are shown in Figures 11–14 and Table 1. The comparison shows that the proposed algorithm achieves the best performance among these real-time trackers. The proposed tracker achieves a success-rate AUC score of 0.64 at real-time speed (123 fps) on Video Sequence I and an AUC score of 0.62 at real-time speed (133 fps) on Video Sequence II.

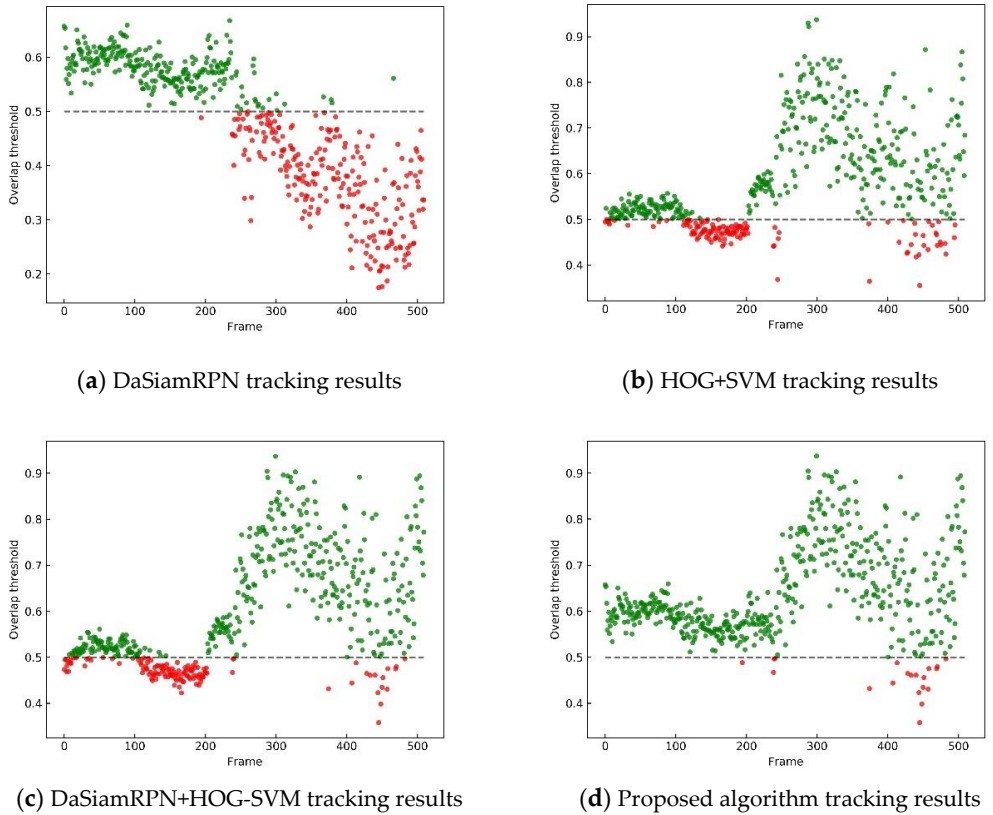

(**a**) DaSiamRPN tracking results　　　　　　　　(**b**) HOG+SVM tracking results

(**c**) DaSiamRPN+HOG-SVM tracking results　　　(**d**) Proposed algorithm tracking results

**Figure 9.** The overlap precision scatter diagram of Video Sequence I with four different trackers.

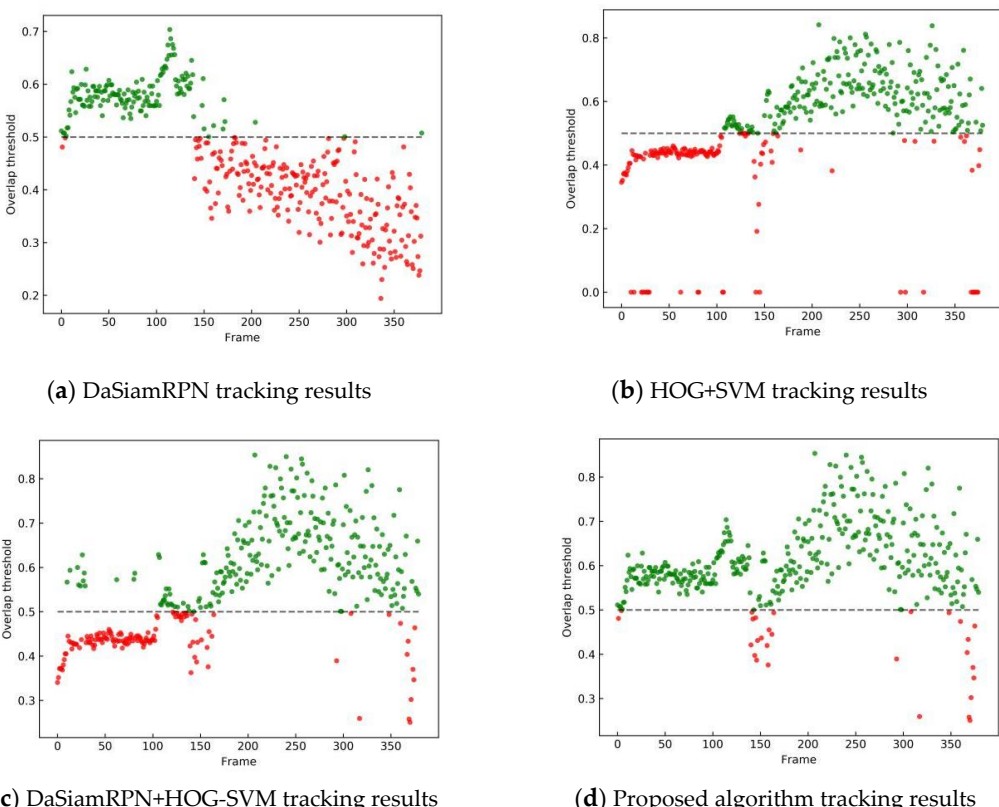

(**a**) DaSiamRPN tracking results　　　　　　　　(**b**) HOG+SVM tracking results

(**c**) DaSiamRPN+HOG-SVM tracking results　　　(**d**) Proposed algorithm tracking results

**Figure 10.** The overlap precision scatter diagram of Video Sequence II with four different trackers.

Compared with DaSiamRPN, HOG-SVM, and DaSiamRPN+HOG-SVM without the key frame detection tracker, our tracker obtains significant success-rate AUC gains of 33.3%, 10.3%, and 6.7% on Video Sequence I, and obtains significant success-rate AUC gains of 29.2%, 17.0%, and 8.8% on Video Sequence II, respectively. Our tracker also generates significant precision AUC gains of 45.8%, 8.2%, and 9.1% on Video Sequence I, and generates significant precision AUC gains of 10.9%, 25.3%, and 10.4% on Video Sequence II, respectively. The results demonstrate that the proposed tracking method is stable and accurate.

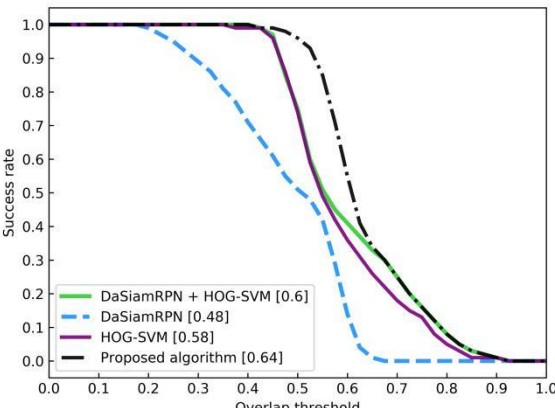

**Figure 11.** The success plots of Video Sequence I.

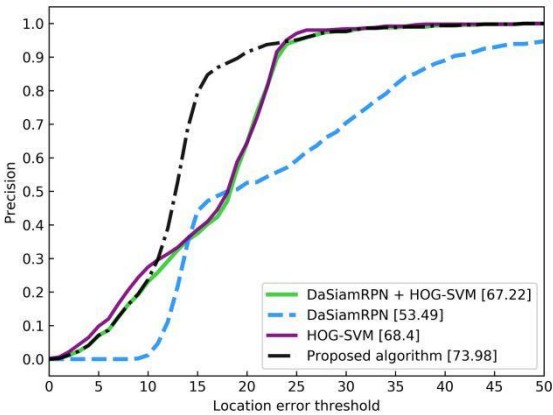

**Figure 12.** The precision plots of Video Sequence I.

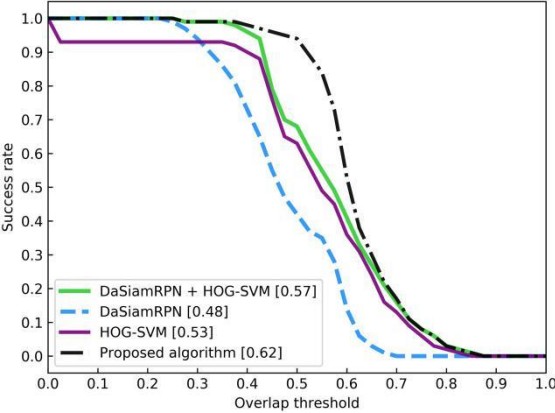

**Figure 13.** The success plots of Video Sequence II.

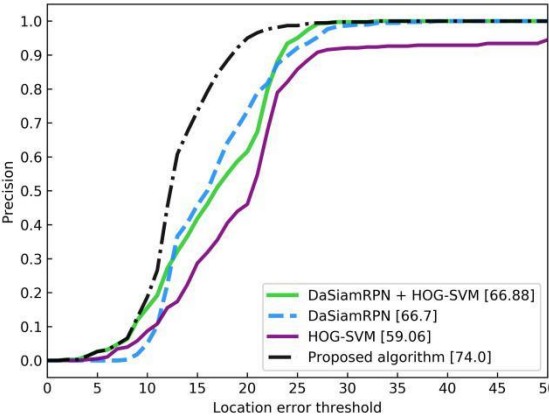

**Figure 14.** The precision plots of Video Sequence II.

As shown in Figure 15, the trackers such as DaSiamRPN, HOG-SVM, and DaSiamRPN+HOG-SVM without the key frame fail to achieve the best performance because they are more vulnerable to inevitable tracking errors, thus corrupting the deep features. In contrast, we involve the Haar feature-based cascade classifier to find the key frame representing dramatic body changes, and the combination method of DaSiamRPN and HOG-SVM is performed after the key frame.

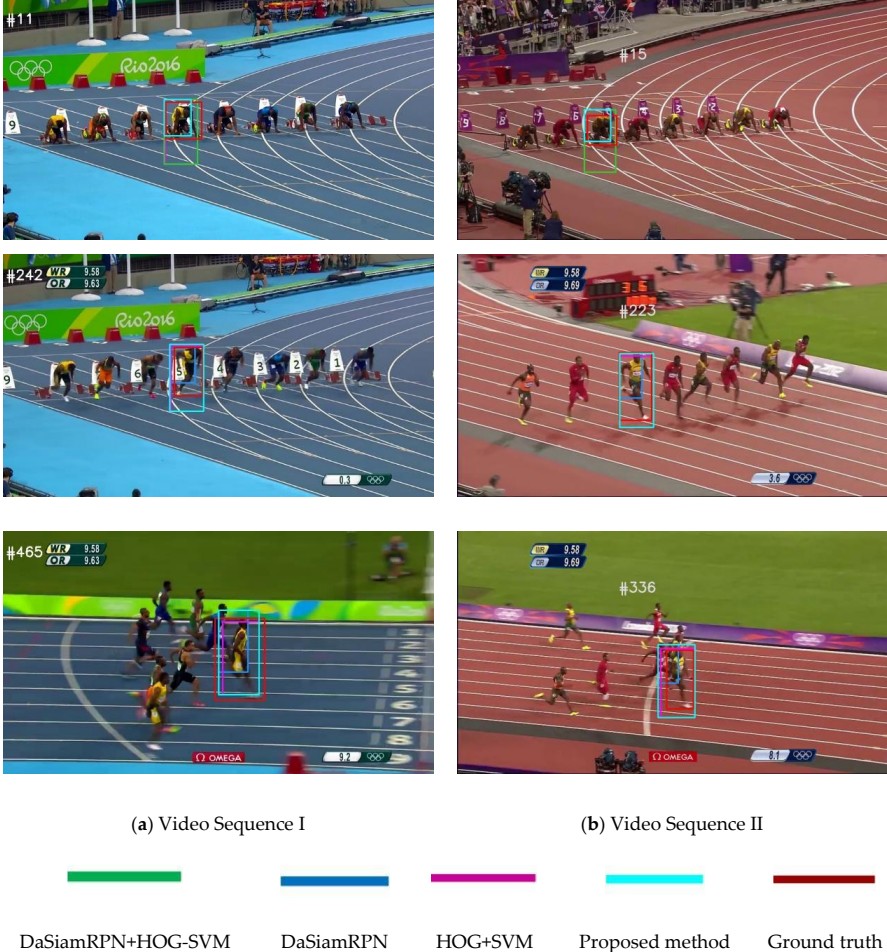

**Figure 15.** Qualitative results of our proposed tracking method, along with DaSiamRPN, HOG-SVM, and DaSiamRPN+HOG-SVM without the key frame. The proposed algorithm accurately and robustly tracks the dramatic changes of the athlete.

**Table 1.** Comparison of the tracking speed (fps) and area under the curve (AUC) with different trackers.

| Dataset | Tracker | AUC↑ | FPS |
|---------|---------|------|-----|
| Video Sequence I | DaSiamRPN | 0.48 | 201 |
| | HOG-SVM | 0.58 | 327 |
| | DaSiamRPN+HOG-SVM | 0.60 | 124 |
| | Ours | 0.64 | 123 |
| Video Sequence II | DaSiamRPN | 0.48 | 195 |
| | HOG-SVM | 0.53 | 324 |
| | DaSiamRPN+HOG-SVM | 0.57 | 135 |
| | Ours | 0.62 | 133 |

## 5. Conclusions

A detection and tracking method combining DaSiamRPN and HOG-SVM with a key frame is proposed for detecting and tracking the athletes in races. The key frame determination method is presented to detect the frames with dramatic body changes. The proposed algorithm consists of two stages depending on the key frame. Before the key frame is found, the DaSiamRPN is selected as the tracking method. In each frame after the key frame, the detection window is enlarged based on the bounding box generated by the DaSiamRPN tracker. In the new detection window, a fusion method, HOG-SVM, is implemented for detecting the athlete, and the tracking results are updated by fusing the tracking results of DaSiamRPN and HOG-SVM in real-time. The experimental results demonstrate that our proposed method achieves state-of-the-art performance on men's 100 m race video sequences. The evaluation results show that the algorithm is stable, accurate, and has excellent real-time performance. Additionally, both the Haar feature-based cascade classifier and HOG-SVM can be directly performed with the OpenCV library. In the future, we plan to focus on the following two directions. First, we plan to construct a data association algorithm to solve the occlusion problem in the race. Second, we plan to improve our method for real-time tracking application in embedded systems of mobile robots.

**Author Contributions:** Conceptualization, methodology, and writing—review and editing, M.S., W.W. (Weizhe Wang), and J.L.; validation and formal analysis, H.Q. and W.W. (Weilin Wan); supervision, project administration, and funding acquisition, L.W. All authors have read and agreed to the published version of the manuscript.

**Funding:** This work is supported in part by the National Natural Science Foundation of China under Grant (51979057, 51609050, 61633009), in part by the Research Fund from Science and Technology on Underwater Vehicle Technology (6142215180209), in part by the Defense Industrial Technology Development Program (JCKYS2019604SXJQR-09), and in part by the Fundamental Research Funds for the Central Universities Facing International Academic Frontier Support Program (3072019CFG0101).

**Conflicts of Interest:** The authors declare no conflict of interest.

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
