# Peer review of "A Novel Changing Athlete Body Real-Time Visual Tracking Algorithm Based on Distractor-Aware SiamRPN and HOG-SVM"

_electronics, doi:10.3390/electronics9020378_

Round 1
Reviewer 1 Report
The topic is interesting. The abrupt changes of position and location of an athlete are indeed a problem in their real-time tracking. The proposed solution has potential. I agree the combination of two consecrated methods (DaSiamRPN and HOG-SVM) and detecting key frame representing major changes of athlete coordinate. Very good, but you claim in the title that you watch the athlete in real time and then say that this is a future work. Maybe I didn't get it right. In any case, the proposed method is not strongly supported by experiments. More test bases would have been welcome. Also, detecting the abrupt change of position is somewhat handmade. The movement of an athlete has something brownian in itself and is very difficult to anticipate. The proposed method does not convince me. One last remark: references seem to be set to be many; it seems to me that 40 is a huge number for a journal article.
Author Response
Point 1:Very good, but you claim in the title that you watch the athlete in real time and then say that this is a future work.
Response 1: We are very sorry for our incorrect writing. In the future work section, our intention expression is the application of our method in the unmanned embedded system, so in the end of the conclusions, the last sentence has been corrected as “Second, optimize and improve our method for real-time tracking application in embedded system of mobile robot.”
Point 2: Maybe I didn't get it right. In any case, the proposed method is not strongly supported by experiments. More test bases would have been welcome. Also, detecting the abrupt change of position is somewhat handmade. The movement of an athlete has something brownian in itself and is very difficult to anticipate. The proposed method does not convince me.
Response 2: Considering the Reviewer’s suggestion, we have added a new video sequence including 380 frames, and we annotated each frames and performed our proposed method on it. The new experiments results are demonstrated in Figure 10, 13, 14, 15 and Table 1.
Point 3: One last remark: references seem to be set to be many; it seems to me that 40 is a huge number for a journal article.
Response 3: I agree with the comments, the number of references has been reduced to 32, and only highly relevant references are preserved.

Reviewer 2 Report
The article presents an improved approach to tracking runners during athletics competitions.
I have three main points that should be considered before considering publication:
- please attach the source codes to the developed algorithms and describe what new authors have developed (most of the solutions used/combined by authors are open-source) - I have the impression that mainly parameterization of the final window,
- please compare the results on at least a few sequences, currently there is only one and this is not enough to be sure that the developed method is better than others,
- please clearly state what is a novelty elements of proposed method.
Author Response
Point 1.1: - please attach the source codes to the developed algorithms and describe what new authors have developed (most of the solutions used/combined by authors are open-source)
Response 1.1: We have made a whole procedure of this method and attached it in the end of Section 4. What new we have developed I think include three main contributions as follow:
- The dramatic body changes of athlete during the running race has a great influence on athlete tracking. DaSiamRPN has achieved a leading position in real-time evaluation, but it heavily relies on the given position in the initial frame. In order to reduce the influence of body changes on tracking results, DaSiamRPN is involved for generating a based bounding box. Therefore, a combined tracking method of DaSiamRPN and HOG-SVM based on key frame is proposed to detect and track the athletes during the race.
- A cascade classifier based on Haar feature is implemented for catching the convert key frame. During the running race, the height and width of bounding box of athletes identified by cascade classifier are calculated to determine the key frame which representing the frame with athlete body changing the most dramatic. This method not only helps to reduce the tracking error by DaSiamRPN, but also helps switch to a new tracking mode.
- We propose a novel tracking method utilizing the DaSiamRPN and HOG-SVM athlete tracking results at the same time. Before the application of HOG-SVM ,we have trained thousands of positive and negative samples of athletes. The proposed algorithm can further enhance the adaptability of athlete body dramatic change. Experiments on actual race video sequences indicate that our proposed algorithm can realize accurate tracking of athlete in real-time.
Point 1.2: - I have the impression that mainly parameterization of the final window
Response 1.2: As the final tracking result, we utilize a bounding box to represent the final window, and we explain it more clearly in Section 3.4.
Point 2:- please compare the results on at least a few sequences, currently there is only one and this is not enough to be sure that the developed method is better than others
Response 2: Considering the Reviewer’s suggestion, we have added a new video sequence including 380 frames, and we annotated each frames and validated our proposed method on the new sequence. The experiments results are demonstrated in Figure 10, 13, 14, 15 and Table 1.
Point 3: - please clearly state what is a novelty elements of proposed method
Response 3: Considering the Reviewer’s suggestion, we clearly state a novelty elements of proposed method including three main contributions in the end of Section 1.

Round 2
Reviewer 1 Report
The paper is somewhat better written. Unfortunately, you still play with the words (for example "optimize and improve our method"). Then, you confuse simulation with real time. May be I am wrong, but I still think the title is inappropriate. Anyway, the paper can be accepted for publication. However, I would like to be removed from the text the word "optimize".
Author Response
Point 1:The paper is somewhat better written. Unfortunately, you still play with the words (for example "optimize and improve our method"). Then, you confuse simulation with real time. May be I am wrong, but I still think the title is inappropriate. Anyway, the paper can be accepted for publication. However, I would like to be removed from the text the word "optimize".
Response 1: We are very sorry for our incorrect writing. Considering the Reviewer’s suggestion, the word "optimize" has been removed from the text, and the last sentence in section 5 has been corrected as “Second, improve our method for real-time tracking application in embedded system of mobile robot.”

Reviewer 2 Report
I agree for publication of improved manuscript.
Author Response
Point 1:I agree for publication of improved manuscript.
Response1 : Thank you for your affirmation of our improved manuscript.
